Genetic diversity and demography of Bufo japonicus and B. torrenticola (Amphibia: Anura: Bufonidae) influenced by the Quaternary climate

Fukutani Kazumi 1 fukutani.kazumi.55a@st.kyoto-u.ac.jp
Matsui Masafumi 1
Tran Dung Van 2 3
http://orcid.org/0000-0002-6274-4959 Nishikawa Kanto 1 2
1 Graduate School of Human and Environmental Studies, Kyoto University , Kyoto , Japan
2 Graduate School of Global Environmental Studies, Kyoto University , Kyoto , Japan
3 Wildlife Department, Vietnam National University of Forestry , Hanoi , Vietnam
Sosa Victoria
Electronic publication date: 2022 Jun 8
Publication date: 2022
Volume: 10
Electronic Location ID: e13452
Received 2022 Jan 25; Accepted 2022 Apr 27
Copyright: © 2022 Fukutani et al.
Copyright year: 2022
Copyright holder: Fukutani et al.
License: This is an open access article distributed under the terms of the Creative Commons Attribution License, which permits unrestricted use, distribution, reproduction and adaptation in any medium and for any purpose provided that it is properly attributed. For attribution, the original author(s), title, publication source (PeerJ) and either DOI or URL of the article must be cited.
License URL: https://creativecommons.org/licenses/by/4.0/

Keywords: Calibration of demographic transition, Ecological niche models, Endemism, Japanese amphibians, Last glacial maximum, Refugia

Funding: JSPS KAKENHI JP21J15839 Environment Research and Technology Development Fund JPMEERF20204002 Environmental Restoration and Conservation Agency of Japan Sasakawa Scientific Research Grant from the Japan Science Society This research was supported by a JSPS KAKENHI Grant (JP21J15839), the Environment Research and Technology Development Fund (JPMEERF20204002) of the Environmental Restoration and Conservation Agency of Japan, and the Sasakawa Scientific Research Grant from the Japan Science Society. The funders had no role in study design, data collection and analysis, decision to publish, or preparation of the manuscript.

==============================
The Quaternary climate affected the present species richness and geographic distribution patterns of amphibians by limiting their activities during the glacial period. The present study examined the phylogenetic relationships of Japanese toads (Bufo japonicus and B. torrenticola) and the demography of each lineage from the past to the present based on mitochondrial sequences and ecological niche models. Japanese toads are a monophyletic group with two main clades (clades A and B). Clade A represents B. j. formosus, including three clades (clades A1, A2, and A3). Clade B contains three clades, two of which corresponded to B. j. japonicus (clades B1 and B2) and the other to B. torrenticola. Clade B2 and B. torrenticola made a sister group, and, thus, B. j. japonicus is paraphyletic. Clades A and B diverged in the late Miocene 5.7 million years ago (Mya) during the period when the Japanese archipelago was constructed. The earliest divergence between the three clades of clade A was estimated at 1.8 Mya. Clades A1 and A2 may have diverged at 0.8 Mya, resulting from the isolation in the multiple different refugia; however, the effects of the glacial climate on the divergence events of clade A3 are unclear. Divergences within clade B occurred from the late Pliocene to the early Pleistocene (3.2–2.2 Mya). Niche similarity between the parapatric clade in clade B (clades B1 and B2) indicated their allopatric divergence. It was suggested that niche segregation between B. japonicus and B. torrenticola contributed to a rapid adaptation of B. torrenticola for lotic breeding. All clade of Japanese toads retreated to each refugium at a low elevation in the glacial period, and effective population sizes increased to construct the current populations after the Last Glacial Maximum. Furthermore, we highlight the areas of climate stability from the last glacial maximum to the present that have served as the refugia of Japanese toads and, thus, affected their present distribution patterns.

Introduction

Biogeographic studies have provided important information on the effects of the Quaternary climate on various species because glacial-interglacial repeated cycles led to their distribution changes, thereby affecting the present distribution (e.g., Taberlet et al., 1998; Hewitt, 2004). Since amphibians are ectotherms and their reproduction is markedly affected by climate factors, they are particularly vulnerable to climate variability (e.g., Carey & Alexander, 2003; Blaustein et al., 2010; Ficetola & Maiorano, 2016). Therefore, the glacial climate had an impact on the present species richness of amphibians by limiting their activities and subsequently restoring the diversity of herpetofauna after the Last Glacial Maximum (LGM; Araújo et al., 2008; Zeisset & Beebee, 2008; Martínez-Monzón et al., 2021).

Japan has rich amphibian fauna with many taxa and high endemism (Nishikawa, 2017). Areas with high species richness may have acted as refugia in the glacial period due to climate stability (Sandel et al., 2011). Furthermore, high endemism may have occurred as a result of in situ diversification by island-specific environments (Kubota, Shiono & Kusumoto, 2015; Kubota et al., 2017). As a result of climate variability in the Quaternary period, multiple refugia for plants, insects, and mammals formed in the Japanese mainland (Hokkaido, Honshu, Shikoku, and Kyushu) during glacial periods, mainly in areas of low elevation, such as coastal areas (e.g., Tomaru et al., 1998; Nunome et al., 2010; Aoki, Kato & Murakami, 2011). Among amphibians widely distributed on the Japanese mainland, the present geographic distribution patterns have been affected by the locations of the refugia, and genetic diversity was increased by isolating to refugia (Tominaga et al., 2013; Dufresnes et al., 2016; Matsui et al., 2019).

In the present study, we focused on the phylogeography of Japanese toads (Genus Bufo Garsault, 1764, Bufonidae Gray, 1825). There are two endemic Bufo species on the Japanese mainland, Bufo japonicus Temminck and Schlegel, 1838 and B. torrenticola Matsui, 1976 (Matsui & Maeda, 2018). Although the effects of the Quaternary climate on European toads have been examined in detail (e.g., Garcia-Porta et al., 2012; Arntzen et al., 2018; Chiocchio et al., 2021), limited information is currently available on B. japonicus and B. torrenticola. Bufo japonicus is widely distributed in Honshu, Shikoku, Kyushu, and some adjacent islands and is a lentic breeder, similar to the majority of congeneric species. This species is divided into two subspecies, B. j. japonicus from western Japan and B. j. formosus Boulenger, 1883 from eastern Japan (Matsui & Maeda, 2018). These two subspecies are distributed parapatrically, and the boundary is in the Kinki region (Matsui & Maeda, 2018). Since the most recent paper proposes raising the two subspecies to the species level (Dufresnes & Litvinchuk, 2021), we herein adopted two subspecies and will discuss the species concept based on nuclear markers in a future study. In contrast to B. japonicus, the range of B. torrenticola is limited to the mountainous areas of central Honshu, with lotic breeding habits unique to Bufo. Bufo torrenticola is distributed overlaying with B. j. formosus and sympatrically in several areas of central Honshu (Matsui & Maeda, 2018). Igawa et al. (2006) previously suggested that geological events during the formation of the Japanese archipelago resulted in genetic diversification in Japanese toads.

Studies combining ecological niche models (ENM) with phylogeography have become mainstream in biogeographic research. The combination of gene-based estimates and analyses of environmental effects provides more robust data (Waltari et al., 2007; Hickerson et al., 2010; Alvarado-Serrano & Knowles, 2014). These methods have helped to resolve the phylogeography of some Japanese anuran species (Komaki et al., 2015; Dufresnes et al., 2016). Since few quaternary fossils of Japanese toads have been found on the Japanese mainland, the combination of genetic and environmental analyses will provide more powerful insights into their Quaternary biogeography. Furthermore, these analyses will help clarify the factors that maintain the high endemism of Japanese amphibians. In the present study, we describe the biogeographic processes that contributed to the diversification of Japanese toads and discuss the effects of the Quaternary climate using genetic analyses and ENM.

Materials and Methods

DNA sampling and sequencing

A total of 213 samples from 191 localities of B. japonicus and 27 samples from 25 localities of B. torrenticola were collected, covering each distribution range (Fig. 1 and Fig. S1; Table S1). The Animal Experimentation Ethics Committee at the Graduate School of Human and Environmental Studies, Kyoto University provided full approval for this research (20-A-5, 20-A-7). We extracted DNA from frozen or ethanol-preserved tissue samples (e.g., muscle, liver, or skin) with the Qiagen DNeasy Blood and Tissue Kit (Qiagen) according to the manufacturer’s instructions.

Figure 1 Phylogenetic relationships and distribution map of Bufo japonicus and B. torrenticola based on mitochondrial cytochrome b haplotypes.

Bootstrap supports (maximum-likelihood)/posterior probabilities (Bayesian inference) are provided for major nodes. Arrows indicate estimated divergence times and 95% HPD (Mya). The scale bar indicates substitutions per site. Enlarged maps with locality numbers are available in Fig. S1, and phylogenetic trees with full haplotype names are available in Fig. S2. The tree was visualized by iTOL v6 (Letunic & Bork, 2021). The map was created by QGIS 3.16 (https://qgis.org). The administrative areas dataset was obtained from the GADM database (www.gadm.org, version 3.4) and the inland water dataset from the Digital Chart of the World available at the DIVA-GIS online resource (www.diva-gis.org). The elevation layer was created by editing the source data from the Geospatial Information Authority of Japan (https://maps.gsi.go.jp/development/ichiran.html).

We amplified mitochondrial DNA from the 3′ region in tRNA-Glu to cytochrome b. We used the newly designed primer set (5′-TTCCTACAAGGACTTTAACCTAGAC-3′; 5′-GTTGGGCTAGTTTGTTCTCTG-3′) for PCR. PCR amplification was conducted in a reaction volume of 10 μl containing 1 μl of 10 × PCR Buffer for Blend Taq (Toyobo, Osaka, Japan), 1.0 μl of the dNTPs mixture (2 mM of each), 0.25 U of Blend Taq DNA polymerase (Toyobo, Osaka, Japan), 0.5 μl of each primer (25 μM), 0.5 μl of the DNA template, and 6.4 μl of distilled water. The PCR protocol followed 2 min of initial denaturation at 94 °C, followed by 33 cycles of denaturation at 94 °C for 15 s, 15 s of annealing at 53 °C, and 90 s of extension at 72 °C, and a final extension at 72 °C for 4 min. Primers, dNTPs, and polymerase were separated from successfully amplified PCR products by precipitation with polyethylene glycol. We performed cycle sequencing reactions (CSR) using the BigDye Terminator v.3.1 Cycle Sequencing Kit (Applied Biosystems, Carlsbad, CA, USA). The same primers used for PCR and two newly designed internal primers (5′-CGAACTTGTTCAATGAATCTGAG-3′, 5′-CTTGTCGAAGTTGGGGTTAAG-3′) were employed for CSR, and the products obtained were purified by ethanol precipitation. Amplified fragments were sequenced on an ABI PRISM 3130 Genetic Analyzer (Applied Biosystems, Carlsbad, CA, USA), assembled with ChromasPro v.1.34 (Technelysium Pty Ltd., South Brisbane, Queensland, Australia), and aligned with MAFFT v7.222 (default parameters: Katoh & Standley, 2013). We aligned 1,071-bp cytochrome b sequences and submitted the haplotypes identified in the present study to the DNA Data Bank of Japan (DDBJ; accession numbers are LC581513–LC581757; Table S1). Cytochrome b regions were extensively used in previous studies on Japanese toads and sufficiently showed variations within Japanese Bufo populations (e.g., Hase, Shimada & Nikoh, 2012; Iwaoka et al., 2021); therefore, we used the same gene fragment to allow for comparisons with previous studies.

Phylogenetic analyses

We built phylogenetic trees using the maximum likelihood (ML) and Bayesian inference (BI) methods. We selected the optimum substitution models for each partition by Kakusan4 (Tanabe, 2011) based on the Akaike information criterion (Akaike, 1974) for the ML analysis and Schwarz’s Bayesian information criterion (Schwarz, 1978) for the BI analysis. The best-fit substitution models chosen for ML and BI analyses were GTR+G models. We performed the ML analysis with estimation node supports by 1,000 bootstrapping replications using RAxML v.8.2 (Stamatakis, 2014). In the BI analysis, we conducted two independent runs of three million generations, each with four Markov chains, and sampled the resulting trees every 100 generations by MrBayes v3.2.6 (Ronquist et al., 2012). We checked the parameter estimates and convergence using Tracer v.1.7 (Rambaut et al., 2018). The initial 10% of trees were discarded as burn-in. Sequences from B. gargarizans gargarizans, B. g. miyakonis, and B. bankorensis were used as outgroups because these sister lineages are the closest relatives of Japanese toads (Matsui, 1984, 1986; Igawa et al., 2006; Table S1).

Divergence dates for Japanese toads were estimated using BEAST v.2.6 (Bouckaert et al., 2019). We used two calibration points, a secondary calibration obtained from a previous study, and fossil evidence. We selected two representative samples from each clade as appeared in our ML phylogeny as described by Garcia-Porta et al. (2012). To introduce calibration points, we added the sequences of four Bufo species and one species belonging to the family Bufonidae as outgroups. The Genbank accession numbers for the outgroups are NC_008410 (B. gargarizans; Cao et al., 2006), NC_027686 (B. stejnegeri; Dong & Yang, 2015), MN432913 (B. bufo; Özdemir et al., 2020), MN432915 (B. verrucosissimus; Özdemir et al., 2020), JN647474 (B. eichwaldi; Recuero et al., 2012), and MT483697 (Epidalea calamita; https://www.ncbi.nlm.nih.gov/nuccore/MT483697).

Two external nodes of Japanese toads were calibrated: (1) the split between B. bufo and B. gargarizans species complexes 12.33 million years ago (Mya; 95% highest posterior density [HPD], 8.81–16.36 Mya) according to the timetree of Garcia-Porta et al. (2012) used a normal prior (mean = 12.3 Ma, standard deviation = 1.93); (2) the oldest fossil record attributable to B. verrucosissimus (1.81–2.59 Mya) used a log-normal prior (offset = 1.81 Ma, mean = 1.0, standard deviation = 0.21) as described by Recuero et al. (2012). The analysis was run for 50 million generations, sampling every 100,000 using the HKY+G model estimated in jModelTest 2.1.10 (Guindon & Gascuel, 2003; Darriba et al., 2012) with the uncorrelated lognormal relaxed clock model. We assessed the stationarity and effective sample size above 200 for all estimated parameters using Tracer v.1.7 (Rambaut et al., 2018). We then generated a maximum clade credibility consensus tree with mean node heights using TreeAnnotator v 2.6 (Bouckaert et al., 2019), discarding the first 10% of the trees as burn-in.

Demographic analyses

Haplotype diversity (Hd) and nucleotide diversity (π) within each main clade were calculated using DnaSP v.6 (Rozas et al., 2017). To examine deviations from neutrality, which are expected with population expansion, we calculated Fu’s FS (Fu, 1997) with 10,000 permutations for significance using Arlequin ver 3.5 (Excoffier & Lischer, 2010). Mismatch distribution analyses were conducted by computing observed pairwise differences to distributions simulated under demographic (Rogers & Harpending, 1992) and range expansion models (Ray, Currat & Excoffier, 2003; Excoffier, 2004) implemented in Arlequin. Observations were compared to model predictions based on 10,000 permutations of data. We also tested the goodness-of-fit of the simulated distribution with the expected distributions using a population expansion model by calculating the sum of the square deviation (SSD). Genetic Landscape Shape interpolation analyses were performed using Alleles In Space (AIS; Miller, 2005; Miller et al., 2006) to obtain spatial patterns in genetic diversity. The analysis produced three-dimensional surface plots of interpolated genetic distances with X and Y coordinates corresponding to geographical locations on the rectangular grid, and surface plot heights (Z) reflecting genetic distances. We performed an analysis of each clade with a grid of 150 × 150 and a distance weighting value of 1.0. All analyses implemented in AIS used sequences as the input matrix (raw genetic distances) and Universal Transverse Mercator coordinates.

We estimated past changes in the effective population size of each lineage in Japanese toads using Bayesian skyline plots (BSP; Drummond et al., 2005) in BEAST v.2.6 (Bouckaert et al., 2019). We employed a calibrated rate for BSP based on the calibration of the demographic transition method (CDT; Hoareau, 2016). This calibration method is an advancement of expansion dating (Crandall et al., 2012) based on the two-epoch demographic model (Shapiro et al., 2004). The commonly used older (>1 Mya) or interspecific phylogenetic calibration often leads to incorrect estimates for intraspecific demographic parameters (Ho & Larson, 2006; Grant, 2015). CDT helps to overcome this issue by using the timing of late glacial climatic warming between 20 and 10 thousand years ago (kya) to calibrate expansion times. We applied CDT on the northernmost lineage following default CDT procedures (Hoareau, 2016) using Beast v1.8.4 (Drummond et al., 2012) because the northernmost lineage was the most likely to be affected by the glacial period and expand during the warming period. We considered the low sample size of the northernmost lineage to have no effect on inferring the past population size because we collected samples to cover their distribution range. BSP analyses were performed for each lineage of Japanese toads using the CDT rate based on the northernmost lineage. We applied the HKY model of molecular evolution as described by Drummond et al., 2005, and a strict molecular clock model for BSP analyses as described by Hoareau (2016). Analyses consisted of one Markov chain Monte Carlo analysis with chain runs for 50 million generations, sampling every 100,000 generations and discarding 10% as burn-in. We verified the effective sample sizes for each parameter and the convergence of chains in Tracer v.1.7 (Rambaut et al., 2018).

ENM

We constructed ENM for each lineage of Japanese toads and predicted their ranges under the present and LGM conditions. We gathered distribution localities with the known occurrence of B. japonicus and B. torrenticola, combining our sampling localities used for phylogenetic analyses in the present study. This initial dataset was filtered to avoid spatial autocorrelation and duplication by randomly selected occurrence points more than 1 km apart from each other in 10 replicates using the R package spThin (Aiello-Lammens et al., 2015). The final dataset comprised 422 records for B. japonicus and 26 records for B. torrenticola, respectively (Table S2). We assigned the records for B. japonicus to the lineages obtained in phylogenetic analyses based on their locations.

We extracted 19 bioclimatic layers representative of the climatic date from 1970 to 2000 from the WorldClim v.2.1 (Fick & Hijmans, 2017), featuring 30 arc seconds of spatial resolutions: 11 layers related to temperature and eight layers related to precipitation. Pearson’s correlation coefficients for all pairs of bioclimatic variables were calculated using ENMTools v.1.4.4 (Warren, Glor & Turelli, 2010) to eliminate predictor collinearity before generating the model. The variables of correlated pairs with |r| > 0.85 were excluded because they were biologically less important based on the known preferences of Japanese toads. The resulting data set contained eight bioclimatic variables: BIO 2 (mean diurnal range), BIO 3 (isothermality; BIO 2/BIO 7), BIO 8 (mean temperature of the wettest quarter), BIO 10 (mean temperature of the warmest quarter), BIO 11 (mean temperature of the coldest quarter), BIO 15 (precipitation seasonality), BIO 18 (precipitation of the warmest quarter), and BIO 19 (precipitation of the coldest quarter).

Distribution models were built with 10 replicates using the default setting in Maxent v.3.4.4 (Phillips, Anderson & Schapire, 2006). We used the areas under the receiving operator characteristics curve (AUC) to evaluate the performance of models. ENM were constructed according to current environmental factors and projected for the present and LGM. To project the ecological niches of Japanese toads on climate conditions in LGM (21,000 years ago), we applied two widely-used general circulation climate models with a 2.5 arc-minute spatial resolution and species-specific masks: the Community Climate System (CCSM4; Gent et al., 2011), and the Model for Interdisciplinary Research on Climate (MIROC-ESM 2010; Watanabe et al., 2011) from WorldClim version 1.4 (https://www.worldclim.org/data/v1.4/worldclim14.html). The logistic thresholds of the 10 percentile training presence generated in Maxent v.3.4.4 (Phillips, Anderson & Schapire, 2006) were used to define the minimum probabilities of suitable habitats.

We also tested niche overlaps among the lineages. We used Schoener’s D (Schoener, 1968) and Hellinger’s I metric (Warren, Glor & Turelli, 2008) to test for niche conservatism and divergence. These metrics were computed from climatic variations under present conditions in ENMTools v.1.4.4 (Warren, Glor & Turelli, 2010). We built niche models of identity and background tests based on 100 pseudoreplicates generated from a random sampling of data points pooled for each pair of clades. Schoener’s D and Hellinger’s I of the true calculated niche between clades were compared with the null distribution by two-tailed t-tests.

We included the putative populations of the introduced origin (see below in Results) for the phylogenetic analysis to identify their haplotypes, but excluded them for the demographic analysis and ENM because they may hinder estimations of the actual demography and suitable distribution area.

Results

Phylogeny and divergence time

Our phylogenetic analyses of mitochondrial cytochrome b (1,071-bp) recovered the monophyly of B. japonicus and B. torrenticola, which diverged from the other Asian Bufo species 7.13 Mya (HPD: 4.31–9.88 Mya; Fig. S3). The monophyly included six mitochondrial clades, five of which corresponded to B. japonicus and the other to B. torrenticola, with varying degrees of divergence (Fig. 1 and Fig. S3). ML and BI phylogenetic trees within Japanese toads were mainly congruent topologies and similar to previous findings (Igawa et al., 2006; Hase, Shimada & Nikoh, 2012). We also resolved possible geographic boundaries between lineages with higher resolution than in previous studies (Fig. 1). Molecular dating estimations revealed that the basal split of Japanese toads between clades A and B occurred 5.66 Mya (HPD: 3.48–8.52 Mya), and the geographic boundary between the two clades was located on the west side of Lake Biwa in the Kinki region (Fig. 1).

The first clade (A) has a wide distribution across the eastern parts of the Japanese mainland and corresponds to B. j. formosus (Fig. 1). This clade is further subdivided into three lineages, which are distributed in the northern Tohoku region (clade A1), from the southern Tohoku to northern Kanto regions (clade A2), and from the southern Tohoku to Kinki regions (clade A3). The common ancestor of clades A1 and A2 diverged from clade A3 1.76 Mya (HPD: 0.85–2.62 Mya), and clades A1 and A2 diverged 0.81 Mya (HPD: 0.34–1.33 Mya). Two samples from Toyama and Ishikawa Prefectures (locality 81, 84; Table S1), which were morphologically identified as B. torrenticola, had the haplotype of clade A3, indicating that the potential genetic introgression of B. j. formosus mtDNA occurred at the boundary between B. j. formosus and B. torrenticola, as suggested in previous studies (Yamazaki et al., 2008; Iwaoka et al., 2021).

The second clade (B) is widely distributed across the western parts of the Japanese mainland (Fig. 1). This clade is subdivided into three lineages: two lineages correspond to B. j. japonicus and one to B. torrenticola. Regarding B. j. japonicus, one lineage is distributed in the Kinki, Chugoku, and Shikoku regions (clade B1), which diverged 3.17 Mya (HPD: 1.83–4.64 Mya), and another in the western end of Honshu to Kyushu (clade B2). The lineage of B. torrenticola is distributed along the mountain range northwest of Lake Biwa and from Hokuriku to the Kii Peninsula. Clade B2 and B. torrenticola made a sister group that diverged 2.25 Mya (HPD: 1.11–3.28 Mya), and, thus, B. j. japonicus is paraphyletic.

Our phylogenetic analysis reconfirmed previously suggested artificially introduced populations in Hokkaido, Izu Islands, and the Kanto region (Matsui, 1984; Kawamura et al., 1990; Igawa et al., 2006; Hase, Shimada & Nikoh, 2012; Matsui & Maeda, 2018; Suzuki et al., 2020; Fig. 1).

Demographic analyses

The high haplotype diversities (Hd = 0.967–0.995), low nucleotide diversities (π = 0.00486–0.00805), and significantly negative Fu’s FS values for all clades of B. japonicus and B. torrenticola supported the pattern of historical demographic expansion (Fu, 1997; Grant & Bowen, 1998; Table 1).

Table 1 Genetic diversity indices and neutrality tests for each clade of Bufo japonicus and B. torrenticola based on cytochrome b sequences.

		N	N a	Hd ± SD	π ± SD	Fu’s FS	
F S	p-value	
Bufo japonicus	clade A1	13	12	0.99 ± 0.04	0.006 ± 0.003	−5.46*	0.0061	
clade A2	33	29	0.99 ± 0.01	0.006 ± 0.003	−24.10*	0	
clade A3	83	57	0.98 ± 0.01	0.007 ± 0.004	−24.83*	0	
clade B1	45	36	0.99 ± 0.01	0.008 ± 0.004	−23.06*	0	
clade B2	28	26	0.99 ± 0.01	0.005 ± 0.003	−24.34*	0	
Bufo torrenticola	25	19	0.97 ± 0.02	0.006 ± 0.003	−8.15*	0.0018	
Note:

N, number of individuals; Na, number of haplotypes; Hd, haplotype diversity; π, nucleotide diversity; SD, standard deviation. Asterisks indicate significant p-values (p < 0.01).

We demonstrated possible patterns of demographic expansions, which indicated the presence of glacial refugia at LGM. The ragged mismatch distribution for clade B1 suggested demographic equilibrium, whereas the unimodal distribution for clades A1, A2, and B2 indicated recent population expansion (Harpending, 1994; Fig. 2). Two peaks for clade A3 and B. torrenticola suggested the inclusion of multiple populations, each undergoing bottlenecks followed by expansion (Hayes et al., 2008). SSD values did not reject the demographic expansion of clades A1 and A3 or the spatial expansion of clades A1, A3, and B. torrenticola (p > 0.05; Fig. 2).

Figure 2 Demographic analyses of each clade of Bufo japonicus and B. torrenticola based on mitochondrial sequencing data.

Left charts display the distribution of observed (histograms) and expected (orange solid lines: under demographic expansion, and green solid lines: under spatial expansion models) pairwise nucleotide differences. The sums of squared deviations (SSD) and p-values are shown for demographic and spatial expansion models. Asterisks indicate significant p-values (p < 0.05). Right charts display Bayesian skyline plots (BSP) showing the evolution of an effective population size (Ne) over time (blue solid lines: median estimates, and blue dashed lines: 95% confidence intervals of highest posterior densities). Vertical lines show the time to the most recent common ancestor (solid lines: median, and dotted lines: lower estimates).

Genetic Landscape Shape interpolation analyses revealed the geographic gradient of genetic variation in each clade (Fig. 3). We show areas with higher genetic diversity in warmer colors and those with lower genetic diversity in cooler colors. High genetic diversity areas for clade A1 were distributed in the southern and western ranges, while those for clade A2 had higher genetic diversity in the southern range. The areas with high genetic diversity in clade A3 were distributed in areas of low elevation on both sides of the Japan Alps (Hida, Kiso, and Akaishi Mountains) at the center of Honshu. Clade B1 had high genetic diversity in the western side of their distribution and in the central areas of the Chugoku and Kinki regions, and clade B2 had high genetic diversity in the northern region. Bufo torrenticola had high genetic diversity, mainly in the southern area and scattered northeastern, northwestern, and central regions. Since populations that remained in refugia during the glacial period have a longer dynamic history and greater genetic diversity than those that expanded after the glacial age (Comes & Kadereit, 1998; Taberlet et al., 1998), regions with high genetic diversity may be regarded as refugia.

Figure 3 Results of Genetic Landscape Shape interpolation analyses of each clade of Bufo japonicus and B. torrenticola.

The geographic distribution patterns of genetic diversity are shown for each clade. Areas with higher genetic diversity are shown in warmer colors, and those with lower genetic diversity are shown in cooler colors. Open circles indicate the localities of samples used for Genetic Landscape Shape interpolation analyses. Maps were created by QGIS 3.16 (https://qgis.org).

The CDT rate based on the northernmost lineage (clade A1) was high at 0.166 changes/site/million years, but was consistent with the findings of Hoareau (2016) and other evolutionary rates estimated for a recent time scale for mitochondrial cytochrome b (Ho et al., 2005; Suzuki et al., 2015). BSP reconstructed the demographic histories of the mtDNA lineages of Japanese toads from the most recent common ancestor (Fig. 2). All of the lineages presented signals of population expansion. Population expansion occurred at different times: between 20 and 10 kya, used for CDT, in clades A1, A2, and B2, before 20 kya in clade B2, and after 10 kya in clade A3 and B. torrenticola. Increases in the effective population size (Ne) were larger (more than a 10-fold increase) for B. japonicus (clades A and B) than for B. torrenticola (less than a 10-fold increase; Fig. 2).

ENM

Each ENM estimated under current climate conditions had mean test AUC values ≥0.9, indicating a better than random prediction. The predicted potential niche models for each lineage of Japanese toads under the climate conditions in LGM are shown in Fig. 4, and those under the present climate conditions are shown in Fig. S4.

Figure 4 Predicted suitable distributions under the last glacial maximum (LGM; CCSM and MIROC scenarios) for Bufo japonicus and B. torrenticola.

Warmer colors indicate higher probabilities of occurrence. Navy blue zones, land areas at LGM; grey zones, oceanic areas; white lines, the present land areas. Maps were created using R package map data version 2.3.0 (Becker, Wilks & Brownrigg, 2018).

The extent of suitable range in LGM in clade A varied depending on the global circulation model. Predicted distributions showed that the suitable range in LGM for clade A1 almost vanished from all areas based on MIROC, while some small parts of coastal areas by the Japan Sea remained based on CCSM. According to the CCSM model, suitable environmental conditions in LGM for clade A2 were distributed in some areas along the coast of the Sea of Japan and the Pacific Ocean, whereas based on MIROC, suitable conditions were expanded distributed along the Pacific coast from southern Tohoku to Shikoku. Regarding clade A3, the predicted suitable distribution range in LGM mainly expanded from Chubu and Kinki according to the CCSM and MIROC models. On the other hand, the CCSM and MIROC models for each clade in clade B both suggested that the projected potential niche models for LGM were significantly limited southward of their ranges.

Niche overlap under the present climate conditions between lineages ranged between 0.04 and 0.59 for Schoener’s D and between 0.18 and 0.85 for Hellinger’s I metrics (Table 2). The null hypotheses of the niche identity test were rejected for all pairs of lineages (p < 2.2E−16), indicating that the environmental niches of the all pairs were not equivalent.

Table 2 Niche similarity scores of Schoener’s D (above the diagonal) and Hellinger’s I (below the diagonal) obtained from known occurrences between lineages of Bufo japonicus and B. torrenticola.

	A1	A2	A3	B1	B2	B. torrenticola	
A1	-	0.45	0.26	0.22	0.04	0.20	
A2	0.72	-	0.57	0.47	0.16	0.37	
A3	0.52	0.83	-	0.59	0.21	0.48	
B1	0.48	0.77	0.85	-	0.37	0.51	
B2	0.18	0.41	0.49	0.68	-	0.25	
B. torrenticola	0.46	0.69	0.76	0.78	0.53	-	

The null hypotheses of the similarity test were not rejected between clades A1 and A2 based on the direction that tested the known localities of clade A2 to the background range of clade A1 for Schoener’s D and based on both directions for Hellinger’s I (Table 3). Additionally, the null hypotheses of the similarity test were not rejected between clades A2 and A3 based on the direction that tested the known localities of clade A2 to the background range of clade A3 for Hellinger’s I. The observed niche overlaps were significantly higher than expected under the null hypotheses between each pair of B. japonicus (except for between clades A3 and B2 and the not rejected pairs of clades described above) and between clade A1 and B. torrenticola, indicating that each lineage was more similar than expected (Table 3). The contrasting results of the identity test and similarity test are false positive; the identity test is more likely to unduly reject the null hypothesis of niche identity (Peterson, 2011). In addition, the background test is known to be more suitable for understanding speciation than the identity test (Smith & Donoghue, 2010). Therefore, we focused on the similarity test, similar to Collart et al. (2021), because the null hypotheses of identity tests were rejected for all of the lineages in the present study.

Table 3 Results of background similarity tests.

The t- and p-values in two-tailed t-tests and whether the observed niche similarities are more or less similar than expected by chance (p < 0.01) are shown.

			Lineage for the background distribution	
A1	A2	A3	
t-value	p-value	similarity	t-value	p-value	similarity	t-value	p-value	similarity	
lineage for the observed distribution	Schoener’s D	A1	-	-	-	−4.06	1.E−04	more	−8.48	2.E−13	more	
A2	0.36	0.72	NS	-	-	-	−3.43	9.E−04	more	
A3	−20.32	2.E−16	more	−8.69	8.E−14	more	-	-	-	
B1	−15.20	2.E−16	more	−837.44	2.E−16	more	−29.88	2.E−16	more	
B2	−32.92	2.E−16	more	−29.73	2.E−16	more	−13.25	2.E−16	more	
B. torrenticola	−26.40	2.E−16	more	−23.64	2.E−16	more	−69.19	2.E−16	more	
Hellinger’s I	A1	-	-	-	−1.10	0.27	NS	−3.20	2.E−03	more	
A2	−1.44	0.15	NS	-	-	-	1.52	0.13	NS	
A3	−16.50	2.E−16	more	−5.79	8.E−08	more	-	-	-	
B1	−15.08	2.E−16	more	−434.87	2.E−16	more	−25.06	2.E−16	more	
B2	−37.04	2.E−16	more	−27.28	2.E−16	more	−10.02	2.E−16	more	
B. torrenticola	−22.78	2.E−16	more	−20.86	2.E−16	more	−56.47	2.E−16	more	
			Lineage for the background distribution	
B1	B2	B. torrenticola	
t-value	p-value	similarity	t-value	p-value	similarity	t-value	p-value	similarity	
lineage for the observed distribution	Schoener’s D	A1	−43.23	2.E−16	more	32.67	2.E−16	more	−17.43	2.E−16	more	
A2	−37.68	2.E−16	more	−5.29	7.E−07	more	15.41	2.E−16	less	
A3	−20.50	2.E−16	more	13.92	2.E−16	less	31.92	2.E−16	less	
B1	-	-	-	−15.64	2.E−16	more	53.89	2.E−16	less	
B2	−7.63	1.E−11	more	-	-	-	4.45	2.E−05	less	
B. torrenticola	−36.69	2.E−16	more	−24.64	2.E−16	more	-	-	-	
Hellinger’s I	A1	−38.55	2.E−16	more	−19.32	2.E−16	more	−20.14	2.E−16	more	
A2	−30.82	2.E−16	more	−7.38	5.E−11	more	7.02	3.E−10	less	
A3	−19.02	2.E−16	more	12.98	2.E−16	less	35.61	2.E−16	less	
B1	-	-	-	−18.25	2.E−16	more	63.10	2.E−16	less	
B2	−4.01	1.E−04	more	-	-	-	10.53	2.E−16	less	
B. torrenticola	−30.27	2.E−16	more	−25.15	2.E−16	more	-	-	-	

The environmental niches of B. japonicus (except for clade A1) and B. torrenticola were more similar than expected based on the habitat available to B. japonicus, but diverged more than expected based on the habitat available to B. torrenticola (Table 3). These contrasting results were also confirmed between clades A3 and B2. This counterintuitive result is likely to be driven by differences in the heterogeneity of the environmental background for the two species (Nakazato, Warren & Moyle, 2010), and their overall similarity was indicated to be low.

Discussion

Phylogeography of Japanese toads

The divergence time between clades A and B (5.7 Mya; Fig. 1) fell within the timeframe reported for other Japanese frogs (7–5 Mya; Nishizawa et al., 2011; Dufresnes et al., 2016). The ancient basins, described as a divergence factor in a previous study (Igawa et al., 2006), were dammed in the middle Miocene under warm and humid climates by the strength of the East Asia summer monsoon (Hatano & Yoshida, 2017). These dammed ancient basins were likely to limit the route between eastern and western Japan. In addition to the ancient basins, global cooling in the late Miocene, related to an intensified East Asian winter monsoon (Herbert et al., 2016; Matsuzaki, Suzuki & Tada, 2020), may also have restricted the activities of frogs. Japanese frogs may have diverged into eastern and western populations by being allopatrically isolated.

The divergence pattern and time within clade A are equivalent to those of the northern lineages of Cynops pyrrhogaster, a lentic breeder similar to B. japonicus, which diverged with glacial cycles (Tominaga et al., 2013). The dry climate at LGM may have affected lentic-breeding amphibians by limiting breeding places.

The results of ENM recognized suitable areas in CCSM for clade A1 in LGM along the Japan Sea coast in the northern Tohoku region, consisting of a region with high genetic diversity (Figs. 3 and 4). We also found a high genetic diversity area of clade A1 in the southern part of the distribution; however, this southern area was not suitable in LGM. The southeastern area of the present distribution of clade A1 on the Pacific Ocean side was also an unsuitable area despite the actual distribution (Fig. S4). There may have been areas that were not suitable for clade A1 based on climate factors, but were inhabitable. The suitable areas during LGM for clade A2 varied between CCSM and MIROC; however, judging from the area with high genetic diversity, the refugia for clade A2 might be along the Pacific coast in the southern Tohoku region. The divergence time between clades A1 and A2 (0.8 Mya; Fig. 1) fell within the middle Pleistocene transition when glacial cooling became severer (1.25–0.7 Mya; Lisiecki & Raymo, 2005; Clark et al., 2006) and a significant flora change also occurred on the Japanese mainland (Momohara, 2016). By assuming that refugia in the glacial age before LGM during the Quaternary climate were consistent with those in LGM, clades A1 and A2 may have diverged by isolation into different refugia along the coastal areas of the Japan Sea and the Pacific Ocean, respectively, followed by genetic drift (Provan & Bennett, 2008).

Although refugia in slightly different locations may have been sufficient to allow divergence, we cannot conclude that a common ancestor of clades A1 and A2 diverged from clade A3 in different refugia because the refugia of clades A2 and A3 were located close to each other.

Regarding the taxonomic treatment of clade B, the genetic diversity and niche differences between B. j. japonicus (clades B1 and B2) and B. torrenticola strongly support their distinct species status, in addition to differences in their morphology and breeding behavior. On the other hand, there is sufficient mitochondrial genetic diversity for a heterospecific level between clades B1 and B2. Although niche similarity between clades B1 and B2 suggests a conspecific relationship, in addition to the lack of differences in morphology or breeding behavior, it is not possible to reach a concrete conclusion on their taxonomic statuses without examining reproductive isolation.

Niche similarity between clades B1 and B2 indicate their allopatric divergence (Wiens & Graham, 2005). On the other hand, niche dissimilarity between B. japonicus and B. torrenticola may suggest the possibility of sympatric speciation (Via, 2001); however, we presently have no data for supporting that possibility. Adaptation to the different ecological niches between B. japonicus and B. torrenticola may have allowed B. torrenticola to speciate in a short period (2.2 Myr; Schluter, 2009). Bufo torrenticola and its sister clade, clade B2, are now distributed allopatrically, which may be attributed to the complex phylogeography associated with the formation and transition of the Seto Inland Sea.

Due to the paraphyly of B. j. japonicus between clades B1 and B2, difficulties are associated with estimating the phylogeography within clade B. Paraphyly may result from incomplete lineage sorting caused by recent speciation or ancient hybridization (Maddison, 1997; Funk & Omland, 2003; McKay & Zink, 2010; Toews & Brelsford, 2012). Divergence times within clade B were estimated from the late Pliocene to early Pleistocene (3.2–2.2 Mya); however, these may be overestimation if there was incomplete lineage sorting (Angelis & Reis, 2015) and underestimation if there was gene flow (Leaché et al., 2014). The other limitations are that we set the calibration only on the external nodes of Japanese toads (Ho et al., 2008) and used a single mitochondrial marker; however, divergence times were similar to those for other amphibians distributed in western Japan (Tominaga et al., 2006, 2013; Nishizawa et al., 2011).

Demography from LGM to the present

Clades A1 and A2, distributed in the Tohoku region, shrank their ranges into refugia and expanded after the glacial period. Some amphibians with overlapping distribution with toads also diverged in the Tohoku region (Sumida & Ogata, 1998; Yoshikawa et al., 2008; Aoki, Matsui & Nishikawa, 2013; Tominaga et al., 2013; Yoshikawa & Matsui, 2014; Matsui et al., 2020). Although divergence times did not coincide, the maintenance of genetic structures within the Tohoku region suggests the presence of multiple refugia for amphibians in this region. Amphibians that diverged in the Tohoku region developed tolerance to the cold and may have survived in multiple refugia by moving to areas of lower elevation during glacial periods. On the other hand, some amphibians did not diverge genetically in the Tohoku region (Nishizawa et al., 2011; Matsui et al., 2019). They may have been unable to live in the cold and dry environments of the glacial period, and only had one refugium in the southern region even when there were refugia in the Tohoku region. These differences may reflect current ecological characteristics, such as habitat elevations and breeding seasons.

A region with high genetic diversity for clade A3 was found on both sides of areas of high elevation in central Japan, particularly on the eastern side. This was also demonstrated by the bimodal mismatch distribution, indicating a contemporary geographic barrier to gene flow (Bremer et al., 2005). The central areas were also shown as unsuitable in LGM for clade A3 in ENM. Areas of high elevation in central Japan were covered with glaciers followed by volcanic activity (Ono et al., 2005; Shiba, 2021), which may have prevented clade A3 from expanding its distribution soon after LGM. Even if the population of clade A3 was fragmented, we did not find any phylogroup in the clade, which may have been due to high mobility when their spatial and population expansion, as suggested for many other Bufo species (Yu, Lin & Weng, 2014; Borzée et al., 2017; Dufresnes et al., 2020).

Suitable areas for B. torrenticola in LGM vanished except for the southern end of their distribution (Fig. 4). The narrow suitable habitats at LGM for clades in clade B (including B. torrenticola) may be because they were estimated based only on western Japan’s current temperature and precipitation. If the present habitats are limited more by factors such as interactions with other populations than solely by climate factors, then suitable habitats in LGM may have been underestimated. A Genetic Landscape Shape interpolation analysis suggested that the area with the highest genetic diversity was a southern area; however, genetic diversity was also high in the northern area, and these regions may have become refugia. The lower degree of expansion of the effective population size for B. torrenticola than that for B. japonicus (clades A and B) indicated that B. torrenticola may have been affected less by the glacial climate than B. japonicus, which may have been because lotic environments were more available than lentic environments under the dry climate in the glacial period. Since the northern and southern ends of the distribution became refugia, the undistributed central region with high genetic diversity suggested a separation between the northern and southern populations. Bufo torrenticola and clade A3 had been geographically separated, followed by the expansion after 10 kya, and then, the niche dissimilarity may have enabled their present overlaying.

We identified high genetic diversity for clade B1 on the western side of their distribution and in the central areas of the Chugoku and Kinki regions, although these areas were not identified as a suitable habitat for clade B1. These areas with high genetic diversity coincided with the region of paleo-rivers (Sakaguchi et al., 2021), indicating that clade B1 maintained its population along paleo-rivers.

In contrast to the results of ENM, which showed the almost vanished suitable area with only a few remaining in the southern parts, areas with high genetic diversity were distributed in the northern and southern Kyushu. Volcanic activity in the central Kyushu (Mahony et al., 2011) may have prevented clade B2 from inhabiting this region, which was also suggested by the increase in the effective population size after LGM. Vegetation in central Kyushu was also affected by volcanic activity around LGM, which contributed to the cool climate (Miyabuchi, Sugiyama & Nagaoka, 2012; Miyabuchi & Sugiyama, 2020). Hynobius stejnegeri endemic to Kyushu diverged into northern and southern populations in central Kyushu (as Hynobius yatsui, Sakamoto et al., 2009), indicating that volcanic activity has long restricted amphibian migration.

Refugia have been consistent with stable climate areas since LGM, and frequently harbor highly endemic fauna (Sandel et al., 2011). Climate stability between LGM and the present day has been proposed as a better predictor of species richness in European amphibian species (Araújo et al., 2008). However, Lehtomäki et al. (2018) suggested that climate stability was of relatively minor importance for Japanese amphibians; nevertheless, these findings do not reflect the characteristics of each species. They also indicated that historical climate stability was very important for plant species richness. The identified refugia of Japanese toads appeared to coincide with areas of plant species richness reported in Lehtomäki et al. (2018). Accordingly, the distribution patterns of each lineage of Japanese toads may have been affected by climate stability after expansion from refugia.

Conclusions

The present study on the phylogeography of Japanese toads provides insights into the divergence of each lineage. Most of the divergence times and patterns between lineages were similar to those of other amphibians. Tectonic events during the formation of the Japanese archipelago and the Quaternary glacial-interglacial cycle may have diverged the lineages of Japanese toads in each region. Furthermore, demographic analyses and ENM revealed the localities of refugia that were formed in areas with climate stability, except for a clade influenced by volcanic activity. The present distribution patterns of genetic diversity resulted from expansion from refugia after LGM. Interactions between clades after expansion may also have influenced the current distribution, which will be revealed by examining gene flow in secondary contact zones. Further genome-wide analyses are needed to clarify the evolutionary process of Japanese toads, including the phylogeny within clade B.

Supplemental Information

Supplemental Information 1 Maps of the Japanese mainland (A) and sampling localities of five clades of Bufo japonicus and B. torrenticola in each region (B)–(H).

Clade A1, white circles; clade A2, black circles; clade A3, gray circle; clade B1, black inverted triangle; clade B2, white inverted triangle; B. torrenticola, white diamond. Stars indicate localities with the sympatry of several clades of B. japonicus or B. torrenticola. Asterisks indicate localities with identified introduced haplotypes. Regarding locality information, see Table S1. Maps were created by QGIS 3.16 (https://qgis.org).

Click here for additional data file.

Supplemental Information 2 Phylogenetic relationships within each clade of Japanese toads based on mitochondrial cytochrome b identified in Fig. 1.

Topologies and branch lengths are based on the Maximum Likelihood (ML) tree. Nodal Numbers represent ML bootstrap supports (>70%). The scale bar indicates substitutions per site. Regarding haplotype numbers, refer to Table S1. Trees were visualized by FigTree v.1.4.4 (tree.bio.ed.ac.uk/software/figtree/).

Click here for additional data file.

Supplemental Information 3 Time tree of Japanese toads and outgroups.

Divergence times were estimated with two calibrations. Estimated divergence times and 95% HPD (Mya) are shown around the main nodes. Nodes are placed based on mean node ages. Asterisks represent nodes with posterior probabilities with values over 0.95. Two triangles indicate the calibration points used to obtain the time tree. The scale bar indicates substitutions per site. The tree was visualized by iTOL v6 (Letunic & Bork, 2021).

Click here for additional data file.

Supplemental Information 4 Predicted present suitable distributions for Bufo japonicus and B. torrenticola.

Warmer colors indicate higher probabilities of occurrence. Navy blue zones, land areas; grey zones, oceanic areas. Maps were created using R package map data version 2.3.0 (Becker, Wilks & Brownrigg, 2018).

Click here for additional data file.

Supplemental Information 5 List of samples used to generate the phylogenetic tree of Bufo species with information on vouchers, collection locality, and GenBank accession numbers.

KUHE, Graduate School of Human and Environmental Studies, Kyoto University; TMP, temporary number.

Click here for additional data file.

Supplemental Information 6 Occurrence records used to build ecological niche models for Japanese toads.

We assigned the records for B. japonicus into the lineages obtained in phylogenetic analyses based on their locations.

Click here for additional data file.

Supplemental Information 7 Mitochondrial cytochrome b sequence alignment.

Click here for additional data file.

We acknowledge K. Eto, I. Fukuyama, R. Fukuyama, S. Ikeda, Y. Kawahara, K. Kimura, T. Matsuki, Y. Misawa, S. Mori, T. Shimada, Z. Shimizu, T. Sugahara, T. Sugihara, Y. Tahara, H. Takeuchi, S. Tanabe, A. Tominaga, N. Yoshikawa, and many more collaborators for collecting samples. We also thank our laboratory members for helping with specimen processing and molecular experiments, in addition to three reviewers for their useful comments.

Additional Information and Declarations

Competing Interests

Author Contributions

Animal Ethics

Data Availability

The authors declare that they have no competing interests.

Kazumi Fukutani conceived and designed the experiments, performed the experiments, analyzed the data, prepared figures and/or tables, authored or reviewed drafts of the article, and approved the final draft.

Masafumi Matsui conceived and designed the experiments, authored or reviewed drafts of the article, and approved the final draft.

Dung Van Tran conceived and designed the experiments, analyzed the data, authored or reviewed drafts of the article, and approved the final draft.

Kanto Nishikawa conceived and designed the experiments, authored or reviewed drafts of the article, and approved the final draft.

The following information was supplied relating to ethical approvals (i.e., approving body and any reference numbers):

The Animal Experimentation Ethics Committee in Graduate School of Human and Environmental Studies, Kyoto University provided full approval for this research (20-A-5, 20-A-7).

The following information was supplied regarding data availability:

The raw data is available in the Supplemental Files and at GenBank: LC581513 to LC581757.

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
