# Peer review of "Genetic diversity and demography of Bufo japonicus and B. torrenticola (Amphibia: Anura: Bufonidae) influenced by the Quaternary climate"

_PeerJ, doi:10.7717/peerj.13452_

## Round 0.1 · original submission · Major Revisions

I coincided with comments of the reviewers regarding differences between interpretation of phylogeny and recognition of species. This should be addressed carefully. Also a careful review of the taxonomic treatment should be followed. Some of the suggestions are included in the file that one of the reviewers included. More explanations and justification might be added in relation of considering a single molecular marker.

Reviewer 1 ·

Basic reporting

The ms based on comprehensive samplings on two toads in Japan explored how the climate changes in the region in the past impacted the phylogeographic patterns of the toads. The english is clear. It included sufficinet background. Raw data are shared. And, they proved the hypotheses on diversification.

Experimental design

Eecept only using only one mtDNA marker, the research is meaningful, highlighting the influences by the miocene to pleistocene climate changes on phylogeogrphic profiles of toads in Japan. I have only one question: why you used HKY model for BSP, also with a constant rate?

Validity of the findings

Although they used only one mtDNA marker, the work attempted to supply robust evidences for their conclusions. of course, we are still doutful on the status of B. torrenticola and its closely related clades. Perhaps, in my opinion, in Clade B, there are three species, i.e., clades B1 and B2 represent different species, because they occupied different niche, enough genetic divergence and even no-found morphological differences or behavious divergence. For proving incomplete lineage sorting or gene flow, i think that it is important to using genomic data to detect the possible evens.

Additional comments

Please furhter improve english.

·

Basic reporting

the manuscript is carefully prepared, with a clear and professional English. The literature cited is rich, both concerning the taxa treated and the methodology used. The structure of the manuscript is clear and useful. The choice of supplementary material well done and helpful.
The conclusions follow the given results.

Experimental design

The manuscrpt presents original research based on a well defined questions. It concerns the distribution in space of time of a group of toads that have specific adaptations and distribution patterns. The exploration of these patterns should give inside in the present diversity.

Validity of the findings

The main issue of the manuscript is clearly supported by the data and well argumented.

My main concern goes to the taxonomic treatment of the two species involved.When reading the result section, the discrepancy between the phylogeny presented and the taxonomic interpretation of the Bufo taxa involved in the study seems evident. Then in the discussion section, the problem of the specific status of Bufo torrenticola and a paraphylectic Bufo japonicus are accepted (but see Igawa et al. 2006). I understand that the morphological divergence is more important between Bufo torrenticola and the Bufo japonicus-lineages and that B. torrenticola has adapted to a new niche. Nevertheless, all published phylogenetic analyses show that B. torrenticola is within B. japonicus. This should be set in the discussion section.

Additional comments

As the results and conclusions of this manuscript are based on a rich material, well representing the treated taxa, and on the pertinent statistical analyses, I am in favor to accept this article. The authors might consider the taxonomic aspect I mentioned.

Reviewer 3 ·

Basic reporting

Fukutani et al. reconstructed the phylogeny, estimated the molecular dating, and analysed the demographic of Japanese Bufo from mitochondrial cytb to infer the Quaternary climate-change impact on the focal species. Additionally, the Ecological Niche Modelling data have been used to support the conclusions made from molecular phylogeography analyses. This finding, despite only be inferred from mitochondrial and spatial data, the conclusions can benefit the phylogeography studies of East Asian anurans in general, and the enriched and under-exposed anurans distributed on Japanese archipelago in particular. While the N taxa used for the molecular analyses are robust, and the analyses conducted in this study are adequate to support their conclusions. Despite so, I found the writing in the subtopic related to phylogeny, molecular dating and demographic analysis are not executed well. The scientific terms use such as are not clear, complicated and can be confusing. The problem with clarity is the result of the writing quality itself rather than the data and figures representations. For instance, the phrasing of “identified old but monophyletic radiation” (L 233) is unclear. What does it means by old here, because monophyletic is referring to clade itself, and radiation attributes to the diversity of the clades across time. Another example, “We specified the possible phylogenetic boundaries” (L 236) is confusing and can be simplified to “We resolved the phylogeny of clades”.
Hence, I suggest a correction on clarity of words, and the use of common wordings may help the message to be conveyed to readers.
Another point, the descriptive results/numbers are almost absented in the demographic analysis. Besides that, I also found a problem with consistency from the content structure, from the abstract to results. For instance, the northernmost clades have been emphasised in the abstract. But the northernmost clades of Bufo was missing in the subtopic related to phylogeny in material and methods and it has been missing in the demographic analysis result. Please ensure the essential contents highlighted in abstract are emphasised again in material and methods and results.

Additionally, some rightful citations clearly were missing in the main text, both for taxonomy literatures to support some claims in introduction, and the figure citations in result section.

Please see my all suggestions for improvement relate to these matters in the annotated pdf.

Experimental design

The N taxa to reconstruct the datasets are robust, although the conclusions are heavily relying on cytb fragment, but I found the molecular analyses and ENM conducted in this study are adequate to support their conclusions about the Quaternary climate-change effects on Japanese Bufo phylogeography. No necessity to re-run any analysis or adding any new analysis. Rather, some missing information which are essential to readers such as the preparation of PCR reactions was missing from the main text.

Validity of the findings

The conclusions are heavily relying on cytb fragment, hence it can be the most weakness found in this paper. But I found the molecular analyses and ENM conducted in this study are adequate to support their conclusions about the Quaternary climate-change effects on Japanese Bufo phylogeography. No overclaim in findings, so the authors must clarify in the conclusion section that better approach and better type of data are really needed for future studies.

Additional comments

The general pattern of result is missing in each figure's description. An improvement in description is needed (especially on Fig. 3). Additional proofreading in general can elevate the quality of main text.

Annotated reviews are not available for download in order to protect the identity of reviewers who chose to remain anonymous.

---

## Round 0.2 · Minor Revisions

We appreciate the effort made to improve the paper taking into account issues raised by the two reviewers. However, one of the reviewers suggested few changes to improve clarity of some sections of the article. Please consider them carefully.

Reviewer 3 ·

Basic reporting

Fukutani et al. have addressed most of the critical points required from the first round of revision. I noticed the tremendous improvements made by the authors at improving the clarity of contents, and adding the descriptive results when explaining the molecular divergence and past population dynamics results, as well as improving the general structure of the manuscript. The scentific rebuttal provided by the authors also were acceptable. To increase the clarity in scientific explanations, and for the last round of revision, I suggest the authors to revise the text in the manuscript based on the specific comments I pointed through the annotated pdf.

Experimental design

No comment

Validity of the findings

No comment

Additional comments

Improvement in the abstract and revision on the description of Figure 3 are needed for scientific clarity and also as a selling point for the manuscript.

Annotated reviews are not available for download in order to protect the identity of reviewers who chose to remain anonymous.

---

## Round 0.3 · Minor Revisions

I appreciate the effort made to improve the paper and I am satisfied with the changes made. However, I suggest that your paper be reviewed by a fluent English editor or by a company. I found that the terms used are not the usual for a phylogeography paper (e.g. endowed, or that the Japan archipielago was constructed instead of originated).

---

## Round 0.4 · accepted · Accept

Thanks for correcting the two problems I discovered and for letting me know that the paper was edited by an English editor.